# Ten Years of Provenance Trials and Application of Multivariate Random Forests Predicted the Most Preferable Seed Source for Silviculture of *Abies sachalinensis* in Hokkaido, Japan

**Ikutaro Tsuyama** [1],*, **Wataru Ishizuka** [2] , **Keiko Kitamura** [1], **Haruhiko Taneda** [3]
**and Susumu Goto** [4]

1   Hokkaido Research Center, Forestry and Forest Products Research Institute, 7 Hitsujigaoka, Toyohira, Sapporo, Hokkaido 062-8516, Japan; kitamq@ffpri.affrc.go.jp
2   Forestry Research Institute, Hokkaido Research Organization, Koushunai, Bibai, Hokkaido 079-0198, Japan; wataru.ishi@gmail.com
3   Department of Biological Sciences, Graduate School of Science, The University of Tokyo, 7-3-1, Hongo, Bunkyo, Tokyo 113-0033, Japan; taneda@bs.s.u-tokyo.ac.jp
4   Education and Research Center, The University of Tokyo Forests, Graduate School of Agricultural and Life Sciences, The University of Tokyo, 1-1-1 Yayoi, Bunkyo-ku, Tokyo 113-8657, Japan; gotos@uf.a.u-tokyo.ac.jp
*   Correspondence: itsuyama@affrc.go.jp

**Abstract:** Research highlights: Using 10-year tree height data obtained after planting from the range-wide provenance trials of *Abies sachalinensis*, we constructed multivariate random forests (MRF), a machine learning algorithm, with climatic variables. The constructed MRF enabled prediction of the optimum seed source to achieve good performance in terms of height growth at every planting site on a fine scale. Background and objectives: Because forest tree species are adapted to the local environment, local seeds are empirically considered as the best sources for planting. However, in some cases, local seed sources show lower performance in height growth than that showed by non-local seed sources. Tree improvement programs aim to identify seed sources for obtaining high-quality timber products by performing provenance trials. Materials and methods: Range-wide provenance trials for one of the most important silvicultural species, *Abies sachalinensis*, were established in 1980 at nine transplanting experimental sites. We constructed an MRF to estimate the responses of tree height at 10 years after planting at eight climatic variables at 1 km × 1 km resolution. The model was applied for prediction of tree height throughout Hokkaido Island. Results: Our model showed that four environmental variables were major factors affecting height growth—winter solar radiation, warmth index, maximum snow depth, and spring solar radiation. A tree height prediction map revealed that local seeds showed the best performance except in the southernmost region and several parts of northern regions. Moreover, the map of optimum seed provenance suggested that deployment of distant seed sources can outperform local sources in the southernmost and northern regions. Conclusions: We predicted that local seeds showed optimum growth, whereas non-local seeds had the potential to outperform local seeds in some regions. Several deployment options were proposed to improve tree growth.

**Keywords:** local adaptation; Sakhalin fir; silviculture; seed zone; tree improvement program

## 1. Introduction

Plant species often show local adaptation, which is a process by which populations genetically diverge in response to natural selection specific to their habitat [1]. Therefore, maladaptation is often

observed when plants are transplanted to different growth environments [2]. This is also observed for long-lived forest tree species with a wide distribution range which are often genetically adapted to local climatic environments, despite their extensive gene flow [3]. Traditionally, provenance trials of forest trees have aimed to identify optimal seed provenances to ensure successful tree planting [4–6]. These attempts often result in accepting local seeds to avoid maladaptation caused by environmental mismatch between the afforestation site and the seed origin [7]. In contrast, in some cases, the best performance was achieved by introducing seeds of several species at several planting sites. For example, range-wide provenance trials of *Pinus sylvestris* revealed that progeny derived from warmer climates outgrew local seed sources in central and northern sites, whereas local seeds grew best in southern sites [8].

Range-wide provenance trials are necessary for evaluating the validity of seed zones, choosing appropriate seed sources, and providing transfer guidelines in forest improvement programs [5,7,9–11]. Seed zones are generally established in geo-topographically and climatically distinct regions [12]. Conifers have a long history of provenance trials, such as descriptions of the seed zones for pines in the southern USA based on a series of long-term trials of *Pinus echinata*, *P. elliottii*, *P. palustris*, and *P. taeda* [13]. In British Columbia to Minnesota, seed zones for forestry species have been evaluated and modified based on provenance trials, genotypes, and phenotypes [14]. Seed zones for *P. densifolia* were validated by long-term provenance trials in Japan [15].

While range-wide provenance trials are fundamental for identifying appropriate seed sources for reforestation programs, these trials are costly, time-consuming, and can only handle a limited number of provenances [7]. Recently, application of statistical models has been considered as relevant for establishing seed zones and seed transfer guidelines [11,14,16,17]. One of the pioneering examples is that fine-scale seed transfer guidelines were developed based on multivariate models for white spruce in Alberta, Canada [11]. Recent studies used statistical models (e.g. species distribution models) to predict the potential distribution of forestry species under various climatic conditions [18–21]. These studies suggested solutions for forest conservation management such as future vulnerability of core and buffer conservation areas [22–27]. However, statistical models have not been applied to improve forestry production such as in predicting future growth of plantations. Together with data from provenance trials and environmental factors, statistical models are practical for evaluating seed zones and seed transfer guidelines based on the prediction of traits and local adaptation under given climatic conditions.

*Abies sachalinensis* is a major component of natural forests in Hokkaido, northern Japan. The geographical distribution of *A. sachalinensis* includes Sakhalin, the southern Kuril Islands, and Hokkaido, the northernmost island of the Japanese Archipelago [28]. As one of the most important commercial timber species in Hokkaido, the proportion of timber volume of artificial plantations is approximately 50% and the seedling stock is 25–30%. The breeding program for *A. sachalinensis* began in the 1950s in Hokkaido. During the program, a total of 782 "plus trees", showing good performance in growth and stem straightness, were selected from natural and artificial forests throughout Hokkaido. The initial seed zones were determined in 1985 based on the results of common garden and provenance trials (e.g., [29]) and the local climate. According to the seed zones, breeding programs have been started to establish seed orchards in different regional zones. To improve timber production, these seed zones must be validated. Initial evaluation of seed zones was based on the results obtained from three provenance test sites [30]. For further improvement of future timber production, comprehensive assessment using a range-wide provenance test is necessary to validate the seed zones.

In this study, we applied a statistical model with climatic variables to predict tree height at 10 years after planting of different provenances of *A. sachalinensis* throughout Hokkaido. Furthermore, we proposed appropriate seed sources for achieving the best performance in terms of height growth. Finally, we described modifications to seed zones and seed transfer guidelines.

## 2. Materials and Methods

### 2.1. Study Area

Hokkaido is the northernmost island of the Japanese Archipelago and ranges from N 41°21′–45°3′ to E 139°20′–145°49′ (Figure 1). It has upper temperate forest in the southern peninsula and sub-boreal forest in the northern and eastern parts. Climatic conditions in the western region are affected by the coastal climate of the Sea of Japan, characterized by heavy snowfall in winter (Figure 2). In contrast, the eastern part of the island has cold and dry winters. There are volcanic mountain ranges of approximately 2000 m altitude at the centre of the island, which comprises alpine forests. The natural distribution of *A. sachalinensis* covers most of the montane forests in Hokkaido, where upper temperate to sub-boreal and alpine forests are found.

### 2.2. Regional Groups and Seed Provenances

The different commercial seed zones for *A. sachalinensis* were recently updated by Nakada et al. [31], and five zones were recognized—the West, North, East, Eastern edge, and South. In this study, we evaluated seven regional groups—W, N, EN, EE, ES, S, and SS groups (Table S1, Figure 1). In the East zone, two sub-zones were suggested based on the climatic difference between the Sea of Okhotsk and the Pacific Ocean sides [32]. We adopted this suggestion and subdivided the East zone into the EN and ES groups. Moreover, the Oshima Peninsula in the West and South zones have a warmer climate and different vegetation from other parts of Hokkaido in which wild *A. sachalinensis* is scarcely distributed [33]. Additionally, our previous study revealed that the southern populations showed low genetic diversity and were genetically differentiated from other populations [34]. Thus, we discriminated SS as a regional group referring to the southern part of the South seed zone.

In the 1970s, open pollinated seeds were collected from 88 trees throughout Hokkaido. Most of the selected trees were plus trees. Seeds were distinguished as a "family" based on each mother tree. Meanwhile, selected breeding materials were absent in SS, where open pollinated seeds produced from local trees were bulked and used for planting. Thus, these local seeds were used as a single family. The resulting 89 families and their regional groups are shown in Table S1. The geographical coordinates of the origins of these families were curated from the register books deposited at the Forestry Research Institute, Hokkaido Research Organization (HRO) as provenance locations (Figure 1). Notably, a provenance location of the local family in the SS group was set to the location of a natural stand, assuming the historical origin of the local seeds used. The collected seeds were sown in 1975 in several nurseries in Hokkaido.

### 2.3. Range-Wide Provenance Tests

In 1980, the HRO established range-wide provenance tests at nine localities [35–38]. Test sites were established in seven regional groups including three boundary areas and were managed by code numbers A30 to A38 (Table 1; Figure 1). Six-year-old seedlings were transplanted to these sites in the autumn of 1980. Because of the limited number of seedlings, the number of planted families differed among sites, ranging from 41 to 82 (56 families on average) (Table 1). The average number of planted sites for a single family was 5.7, indicating an effective number of repetitions for the provenance tests.

Three replicates were established at each planting site. Thirty trees per family were planted within each replicate, whereas 40 trees were planted in A35. The tree density ranged from 2200 to 5200 trees per ha according to the conditions of the sites. A complete dataset of tree height and mortality was available until the measurements done in 1989—10 years after transplantation. The measurements at several sites were abandoned after 1990 because of severe meteorological and/or biological damage, which made the range-wide model prediction difficult. We then used the tree height at 10 years after transplantation as a single time point for subsequent analysis.

**Table 1.** Summary of provenance tests for *A. sachalinensis*. The values for tree height and survival rate at 10 years after planting.

| Site Code | Regional Group | Latitude (°N) | Longitude (°E) | No. of Planted Families | Tree Height (m) | Survival Rate (%) |
|---|---|---|---|---|---|---|
| A30 | SS | 41.8488 | 141.1192 | 53 | 3.47 ± 0.70 | 90.5 |
| A31 | W | 42.7481 | 140.6308 | 41 | 2.79 ± 1.41 | 65.6 |
| A32 | S | 42.4586 | 142.4664 | 45 | 3.32 ± 0.96 | 63.7 |
| A33 | W | 43.2803 | 141.8785 | 82 | 3.50 ± 0.90 | 85.7 |
| A34 | N | 44.5550 | 142.2576 | 55 | 1.56 ± 0.75 | 76.8 |
| A35 | EN | 44.2273 | 142.9343 | 51 | 3.17 ± 0.91 | 77.1 |
| A36 | EN | 43.6542 | 143.7846 | 49 | 2.97 ± 0.98 | 59.4 |
| A37 | ES | 42.7309 | 143.4939 | 64 | 3.71 ± 0.84 | 93.7 |
| A38 | EE | 43.0415 | 144.9744 | 64 | 2.37 ± 0.91 | 70.7 |
| Average | | | | 56 | 3.14 ± 1.08 | 75.9 |

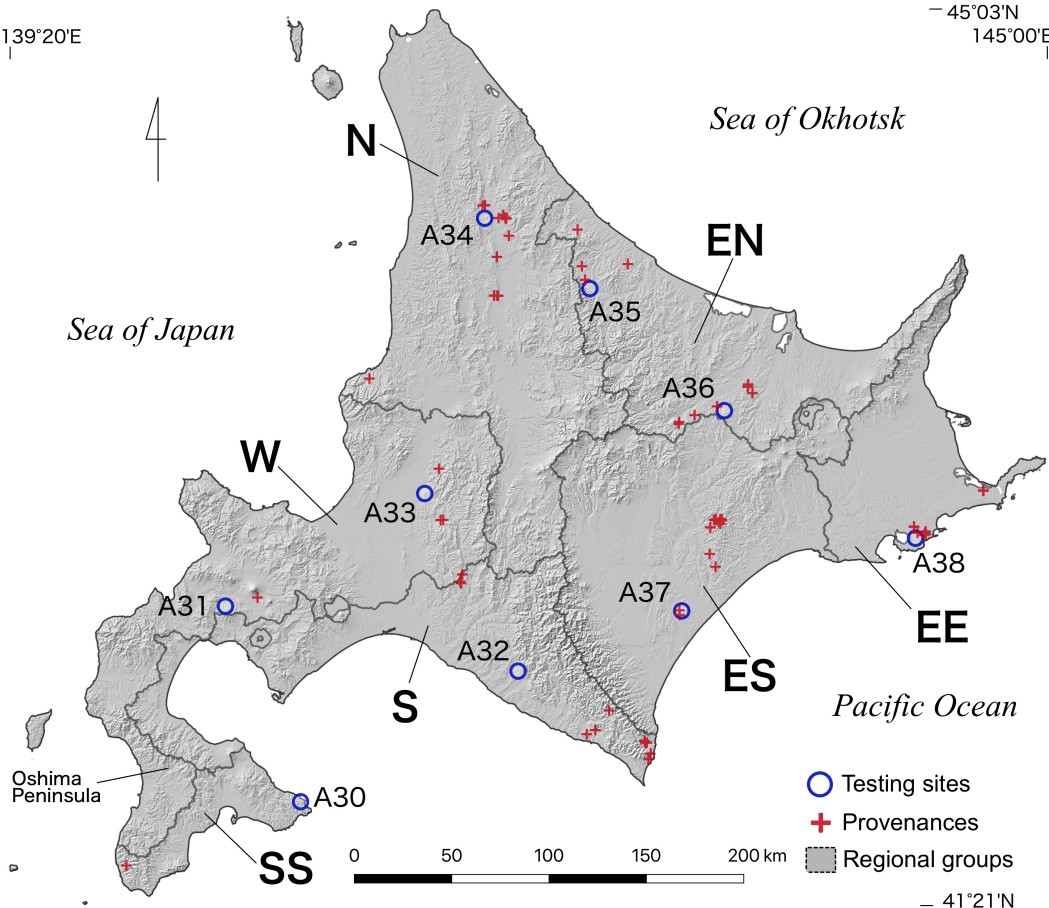

**Figure 1.** A map of study area showing locations of testing sites, provenances, and regional groups.

## 2.4. Climatic Data

We used climatic data for grid cells from the Japan Meteorological Agency [39], which had a spatial resolution of 30″ N × 45″ E (approximately 1 km × 1 km). The following eight climatic variables, which were considered as important for the growth of *A. sachalinensis* according to previous studies [29,32], were calculated (Figure 2). The warmth index (WI) (°C month) was defined as the annual sum of positive differences between the monthly mean temperature and +5 °C [40], indicating an effective heat quantity for the growth of plants. The monthly mean daily minimum temperature of the coldest month (TMC) (°C) provides a measure of extreme coldness. Precipitation during summer

(PRS) (mm) is a sum of precipitation from May to September and represents the water supply during the growing season. The maximum snow depth (MSD) (m) is a measure of snow accumulation. Solar radiation in winter (October to April, WinSR), spring (May, SprSR), summer (June to August, SumSR), and autumn (September, AutSR) (0.1 MJ/m$^2$/day) are measures of energy distributions in each season. Spring, summer, autumn, and winter months for each variable were determined by variables of each month showing mutually positive relationships.

To evaluate the effects of distances in climatic environments between 89 seed provenances (Table S1) and nine testing sites, we calculated the differences in temperatures and solar radiation as well as the relative ratios of precipitation variables including PRS and MSD between the seed provenances and the testing sites. Previous studies also revealed that environmental distances gave better results than the geographic distances for predicting the fitness of other species [2]. Our preliminary analyses used actual climate values as explanatory variables, by which the models did not fit better than the present models (data not shown). Thus, climatic distances between seed sources and planting sites were valid and used as explanatory variables for model construction. Relative ratios were calculated as follows:

$$Rp_i = ((Tp_i - Sp_j)/Sp_j) \times 100 \tag{1}$$

Rp$_i$: relative ratio of precipitation variables (PRS and MSD) between testing site $i$ and seed provenance $j$, Tp$_i$: precipitation variables at testing site $i$, Sp$_j$: precipitation variables at seed provenance $j$.

These climatic distance data were used for model construction. To project the result of prediction throughout the study area, the differences and relative ratios of the climatic variables between each grid cell and mean values among seed provenances in each regional group were calculated.

## 2.5. Statistical Model for the Height Growth of A. sachalinensis

To identify the effects of climatic distances between seed provenances and testing sites on the height growth of *A. sachalinensis*, we constructed multivariate random forests (MRF) [41]. Random forest (RF) is a machine learning method that assembles the results of base learners such as tree-based models with a randomization process that enables high learning performance [42]. MRF extended the RF method to treat unified cases including multivariate response regression. Tree heights at 10 years after planting among seven regional groups were used as response variables (Code S1). The distances of eight climatic variables between seed provenances and testing sites were used as explanatory variables. We applied the MRF to every grid cell throughout Hokkaido and predicted the tree heights in the seven regional groups. Optimum provenances for every grid cell were projected based on the prediction throughout Hokkaido. We excluded the area showing a WI of less than 35 from the projection because these areas are out of the *A. sachalinensis* plantation range (data not shown). "randomForestSRC" package [43,44] on R version 3.6.2 [45] was used to construct the MRF, and QGIS version 3.4 [46] was used for projection.

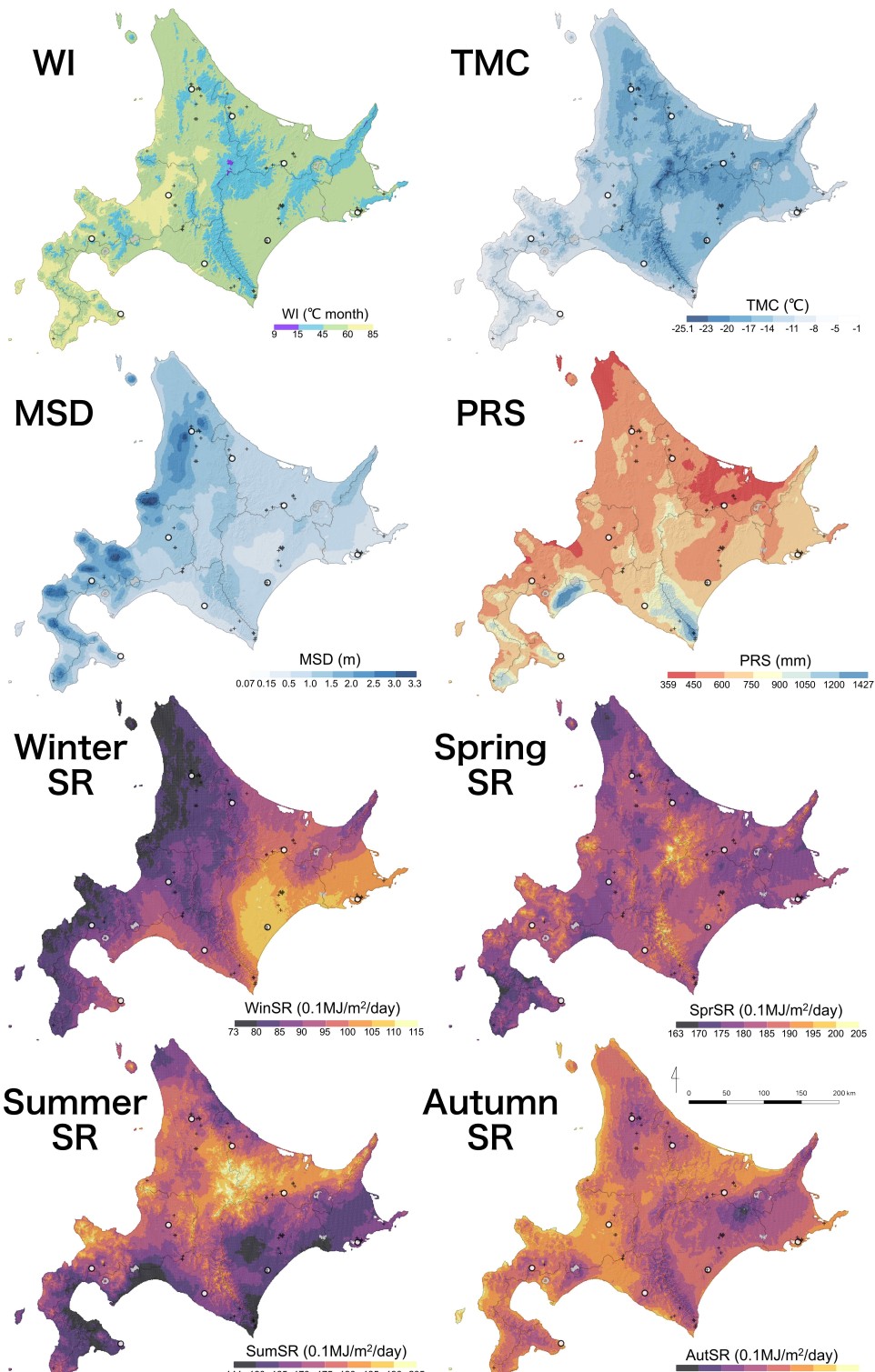

**Figure 2.** Maps for climatic variables used as explanatory variables. Open circles indicate testing sites, and plus signs indicate provenances.

## 3. Results

### 3.1. Measured Height Growth Performance among Testing Sites

The average tree height was 3.16 m among all planting sites (Table 1). The average of tree heights in A37 was 2.4-fold of that in A34. Tree height in A34 was severely compromised by *Scleroderris* canker

disease. For many transplants at this site, a reduced height was frequently observed because of the death of branches by the disease, whereas death of the tree was caused by the disease in a few cases. Tree height was also negatively affected by several meteorological factors at other sites; for example, snow pressure broke branches in A31, late frost damaged young shoots in A36, and winter cold injury or desiccation occurred in A38 where transplants were not covered by snow because of the shallowest snow depth among all testing sites.

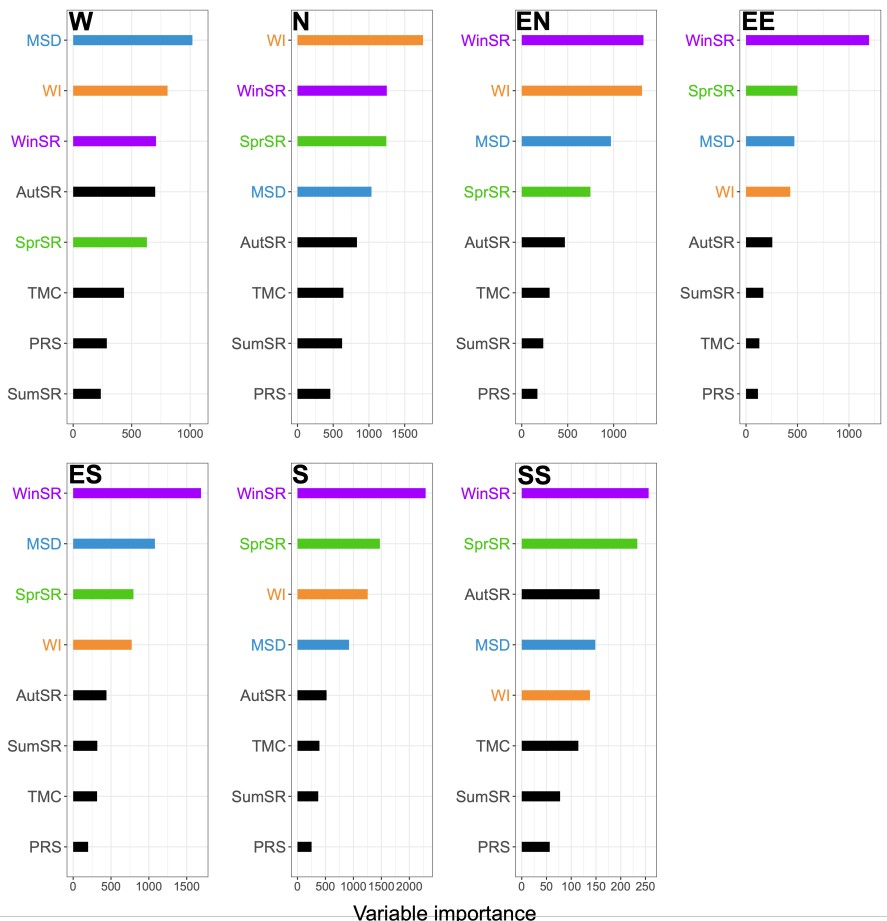

**Figure 3.** Importance of explanatory climatic variables in each regional group in multivariate random forests (MRF). The importance of each variable was identified based on increased mean square errors in the MRF.

*3.2. Model Accuracy and Climatic Conditions Controlling Height Growth of A. sachalinensis*

The MRF for the height growth of *A. sachalinensis* with climatic distances between seed provenances and testing sites showed the following coefficient of determinations ($R^2$)—0.32 for W, 0.36 for N, 0.31 for EN, 0.33 for EE, 0.38 for ES, 0.38 for S, and 0.45 for SS (Figure S1). These $R^2$ values and Figure S1 show that the prediction accuracy of the MRF was not high; however, the overall trends in height growth were reproduced successfully. Variable importance analysis of the regional groups in the MRF showed that WinSR, WI, MSD, and SprSR had relatively important effects on the height growth of *A. sachalinensis*, whereas the order of variable importance differed among the regional groups (Figure 3). WinSR showed the highest importance in most regional groups, except in W and N. MSD showed the highest importance in the W region, which has the heaviest snowfall. In contrast, PRS had the lowest importance, except in the W region.

The response of predicted tree height to the important four climatic variables in the MRF showed that trends in the response to the climatic distances generally differed by region (Figure 4). However, there was a common trend in the responses to WinSR in which the tree height growth exhibited peaks

close to the local point across all regional groups. In addition, two patterns were mainly identified along a geographic gradient; W, N, and SS in the western part of Hokkaido had unimodal peaks in height growth on the positive side of the WinSR distances, whereas EE and ES in the eastern part of Hokkaido retained peaks after the distances when WinSR increased to positive. As for MSD, height growth peaked around zero in the distances (i.e., local environment) among all the regional groups in common. Two patterns in height growth were identified in response to the distances in the MSD; height growth peaked when the distances in MSD decreased to negative values (W, N, and SS) or increased to positive values (EN, EE, ES, and S). The response patterns to distances in the WI and SprSR were consistent among regional groups except SS for WI and W and SS for SprSR, respectively.

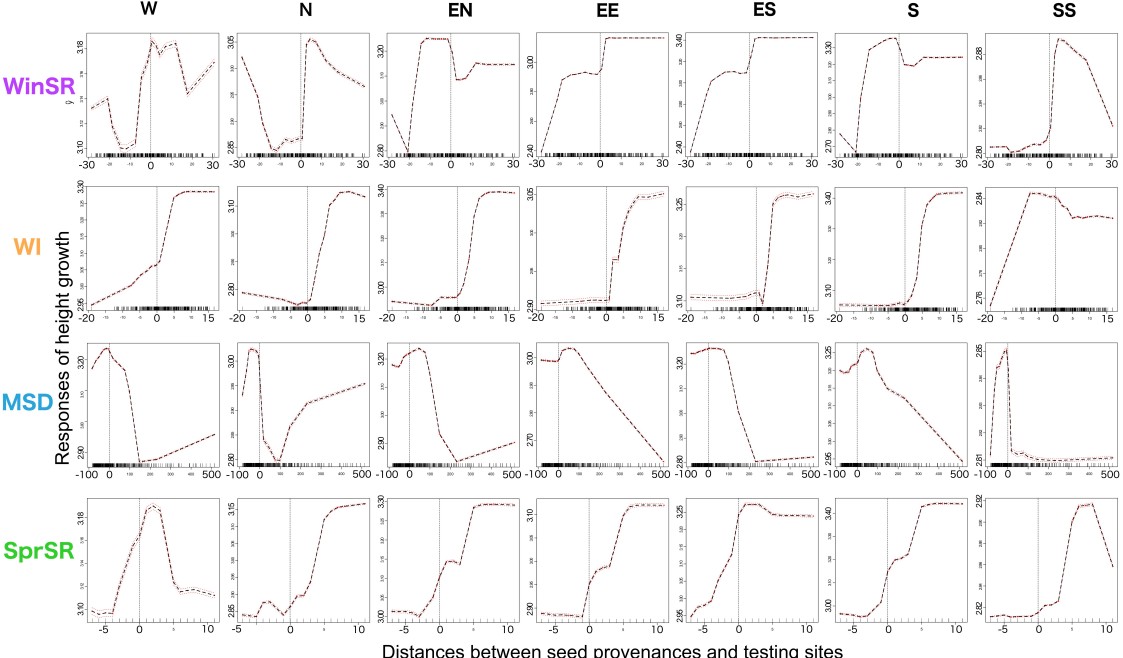

**Figure 4.** Responses of height growth to important climatic variable distances in each regional group according to multivariate random forests. Dashed black lines indicate expected values, dashed red lines indicate mean confidence intervals (expected values $\pm$ 2 $\times$ standard errors), and rugs on x-axes indicate data existence.

### 3.3. Prediction Maps for Tree Height

Assuming that local seeds were planted in each regional group, we predicted the tree height by region based on the prediction of the MRF (Figure 5a). We also projected maps for tree height which assumed that seeds derived from each regional group were planted throughout Hokkaido individually (Figure S2). These maps showed that predicted heights differed among the planted seeds in regional groups because of the difference in the importance and responses to climatic distances (Figures 3 and 4). Trees derived from the ES and S regions grew taller (larger than 3.7 m) in the local region. In contrast, trees were relatively shorter (less than 3.4 m) when they were derived from marginal regional groups (i.e., EE and SS) than those from other groups.

The maximum tree heights for every grid cell were projected to show a potential tree height map which assumed that the best seeds were used for planting, regardless of whether they were local or non-local (Figure 5b). Seeds derived from the ES, S, and EE regions showed similar heights and distribution patterns between local and optimum seed provenance cases (Figures 5a,b and S3). However, in the N and W regions, tree height growth was clearly improved when seeds derived from optimum seed provenances were used.

A map of the optimum regional groups as seed sources for each grid cell showed that seeds derived from local regional groups generally showed the best height growth (Figure 6). The proportions

of local seeds were the highest among the W, EN, EE, and ES regional groups. Particularly, local seeds exhibited the best performance in 95% of the area in S regional group. S was the only regional group selected as an optimum seed source among all regional groups. In the N region, however, the proportions of non-local seeds derived from neighboring regional groups (S, EN, and W) were higher than those of local seeds. In addition, local seeds were typically not selected as the best seed provenance in the SS regional group.

## 4. Discussion

In this study, we successfully estimated the responses of height growth of *A. sachalinensis* to climatic conditions corresponding to seed sources (i.e., regional groups) using MRF, data from range-wide provenance tests, and fine scale (approximately 1 km) climatic data. Prediction maps for the tree height of *A. sachalinensis* were successfully projected by assuming that seeds derived from local or optimum seed sources were used. Moreover, we evaluated the validity of the current framework of seed zones, which used local seed sources and proposed appropriate options to improve the height growth of *A. sachalinensis*.

*4.1. Important Climatic Factors Affecting the Height Growth of* A. sachalinensis

The MRF showed that WinSR, WI, MSD, and SprSR were important factors affecting the height growth of *A. sachalinensis*, although the order of importance differed among regional groups (Figure 3). The model also showed that responses of height growth to climatic variables differed among regional groups, which may reflect different selection regimes to local environments (Figure 4). Genetic differentiation along environmental gradients was circumstantial evidence of local adaptation to the respective environment [34].

For five of the seven regional groups, WinSR was the most important climatic factor. WI and MSD were the most important factors at two sites (N and W, respectively) (Figure 3). The responses to WinSR and MSD showed that the height growth had peaks close to the local environments (Figure 4). These results suggest that WinSR and MSD, climatic factors of the winter, are the main drivers of local adaptation in the height growth of *A. sachalinensis*. Furthermore, these two variables showed another common trend; the response pattern of the height growth to the variables differed between western (W, N, and SS) and eastern regional groups (EE and ES). Hatakeyama [29] also revealed that WinSR and MSD affected height variation among seed provenances. Okada et al. [47] indicated that the number of layers of winter buds, which may be correlated with climatic factors such as WinSR, was significantly different between western and eastern seed provenances in Hokkaido. In addition, our previous study revealed that WinSR was positively related to the genetic differentiation of natural populations along longitudinal gradients, whereas WI, MSD, and SprSR were negatively related to these differences [34]. Therefore, the regional ecotypes of *A. sachalinensis* would be genetically adapted to the local climate along a longitudinal gradient.

In general, the photosynthetic activity of evergreen conifers including *A. sachalinensis* is ceased or extremely low during winter [48]. However, our results suggested WinSR as an important factor affecting the height growth of *A. sachalinensis*. The months of October, November, and April were included in WinSR, in which the photosynthetic activity of *A. sachalinensis* was reported to be active [48]. In addition, some studies suggested that solar radiation in early spring was important for the shoot growth of *A. sachalinensis* and coniferous trees [49,50]. Our results suggest that solar radiation during early winter and spring are important in the local adaptation for height growth of *A. sachalinensis* through photosynthesis, although unmeasured variables, which are highly correlated with winter solar radiation, can be important.

The results of examination of the MSD response revealed sharp phenotypic adaptation in all provenances. Too much snowfall shortens growing season by causing a long snow cover period and increases mechanical damages due to snow pressure [29]; these effects may decrease the height growth of *A. sachalinensis* derived from eastern regions with low snowfall, as these plants

were not adapted to heavy snow. Therefore, MSD is an important selective driver of the local adaptation of *A. sachalinensis* [29]. Ecological divergence at the ecotone between the Sea of Japan side (heavy snowfall) and the Pacific Ocean side (poor snowfall) was also demonstrated in other species such as *Fagus crenata* [51] and *Cryptomeria japonica* [52].

In contrast, PRS was not selected as an important factor in this study (Figure 3). However, PRS is considered as a critical factor determining the distribution of species in the genus *Abies* because *Abies* is generally less drought-resistant than other coniferous species are [53,54]. Our results suggest that the range of PRS in Hokkaido is sufficient for the height growth of *A. sachalinensis* in all regions.

### 4.2. Optimum Regions and Seed Sources for Height Growth of A. sachalinensis

Tree height prediction maps revealed that local seeds generally showed the optimum performance, although the predicted tree heights differed among planting regions (Figures 5 and 6). The height growth of local seeds was fairly good in the western ES and eastern S regional groups (Figure 5a). This suggests that the climatic conditions in these areas are suitable for the height growth of *A. sachalinensis*. In contrast, local seeds did not perform well in the EE region, and even deployment of other seed sources did not improve the performance. These results indicate that climatic conditions in the EE region are unfavorable for *A. sachalinensis* height growth at young ages, although productivity at the mature stage in this region is known to be high. Seed sources were selected as optimum only in the local region except for that in the eastern part of ES, which is adjacent to the EE region. Clear local adaptation to the Pacific Ocean side may be responsible for this trend, which has also been observed for other woody species [51,52].

In some regions including W, N, and SS, seeds from distant sources outperformed those from local sources (Figures 5b and 6). In these regions, tree height growth was clearly improved when non-local but optimum seed sources were used (Figure S3). Particularly, seeds from the S region were the only materials found to be optimum across all regional groups, suggesting that they have high plasticity to grow in a wide range of climatic conditions. The geographic location of the region may be a reason for this; the S region lies on an ecotone between the Sea of Japan side with heavy snow and low solar radiation and the Pacific Ocean side with poor snow and high solar radiation in winter (Figure 2), which is a common characteristic across the Japanese Archipelago. Trees derived from the S region were also known to show intermediate values in some key traits with regional variations, for example freezing tolerance [29,32]. Therefore, seed sources in S may be universal materials useful for wide-range planting (Figure 6, Figure S2). For planting in the N region, seed sources derived from adjacent regional groups were selected as optimum materials as well as local seeds. Unsuitability of distant non-local regional groups seems to be a common pattern of this species, since the consistent result was also recognized in EE and ES regions, even though they had contrasting climatic characters to the N region. Therefore, overall responses to climatic factors were considered to be robust including the N region. However, when considering the optimum materials and their performances in this region, our model seems to have a limit of prediction accuracy. The test site in the N region (A34) severely suffered from *Scleroderris* canker disease. The infection rate was reported to as high as 87.4% until the seventh year after transplantation [38]. The height growth that was affected by this disease was observed to be the smallest in average among all sites, whereas the survival rate (76.5%) was not greatly affected (Table 1). Biological factors, including such diseases, were not incorporated in this study, which may cause decrease in prediction accuracy especially in this region. Further studies are needed to assess the growth potential of local and adjacent non-local materials.

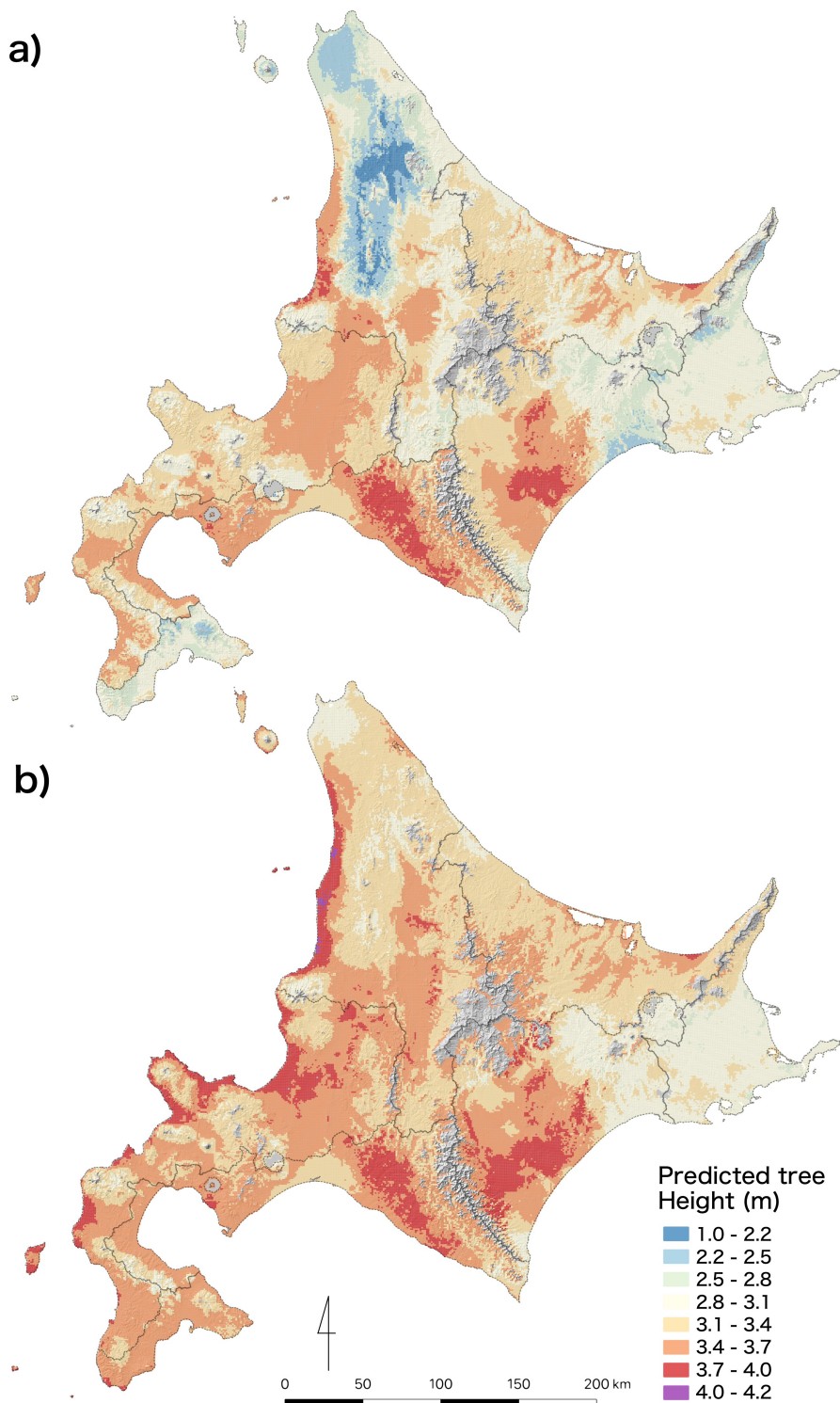

**Figure 5.** Predicted tree height for (**a**) local provenance and (**b**) optimum seed provenance. Gray area indicates alpine zone with less than 35 in the WI, which was excluded from the projection.

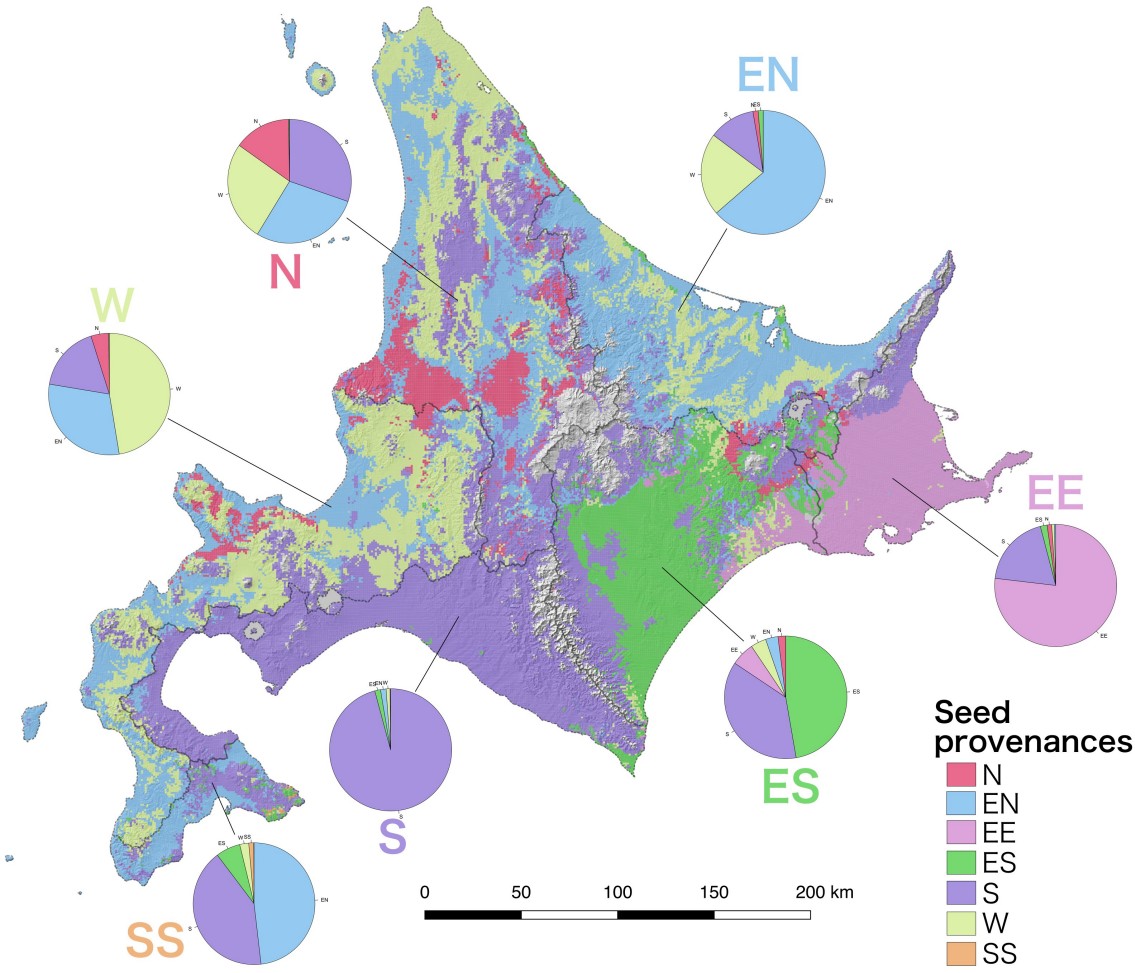

**Figure 6.** Optimum provenance for planting based on predicted tree height by multivariate random forests. Pie charts show the fraction of areas for which each seed provenance was predicted as optimum in each regional group. Gray area indicates alpine zone with less than 35 in the WI which was excluded from the projection.

Similar to the N regional group, some difficulties remain in comprehensively evaluating seed sources. A seed source in the SS regional group was indicated to be an inferior material for local planting. However, further validation of SS is required because only one provenance was evaluated in the present study, and unlike in other regions, the seeds were not derived from selected breeding materials but originated from a local natural stand. Furthermore, the mechanisms of local adaptation of *A. sachalinensis* remain unclear. To achieve successful silviculture, we should consider not only tree growth but also other factors such as mortality [55].

Average 10-year survival rates for each of the regional groups at each testing site showed that the local group or the adjacent groups showed relatively high survival rate in most sites (Figure S4). This pattern was similar to the overall trends shown by MRF in this study (Figures 5 and 6), that is also observed in the averaged tree height in Figure S4. Moreover, our preliminary analysis on mortality showed consistency with the results on tree height in this study, such as common important variables. Therefore, we concluded that the validity of our current results was confirmed, which suggested appropriate deployment of seed sources. It has been widely acceptable for major conifers to use tree height for their evaluation as optimum seed sources, such as *Pinus sylvestris* [56], *P. glauca* [11], and *Picea abies* [57]. Further analysis is also expected to quantify both the growth and survival potentials of each seed sources, since non-local seeds represented complex responses (Figure S4).

*4.3. Implications for Improving the Height Growth of A. sachalinensis*

Range-wide provenance tests have been used to establish and modify seed zones and seed transfer guidelines to improve the tree growth of many forestry species [55,58–60]. There are three conceptual options for improving the current frameworks: (i) re-set the current seed zones, (ii) facilitate appropriate seed transfer, (iii) facilitate assisted gene flow beyond the seed zones to avoid a reduction in tree growth caused by a mismatch to future climatic conditions [7,61,62]. The present study incorporated MRF and can thoroughly propose the former two options for *A. sachalinensis*.

For the first option, our result demonstrated the relevance of the current seed zones of this species, as local seed sources represented optimum growth at most planting sites. Particularly, the validity of local seed sources was clear for the S, ES, and EE regions on the Pacific Ocean side of Hokkaido. In the present study, we set seven regional groups against the current five seed zones based on genetic differentiation in several traits and climatic differences [32]. The East zone was then subdivided into the EN and ES groups between the Sea of Okhotsk and the Pacific Ocean sides. The geographical distribution of the optimum ranges for these two groups clearly indicated that the current inclusive management by one East zone was insufficient (Figure 6) and supported the subdivision of the East zone to use independent seed sources according to climate differences. Because *A. sachalinensis* exhibited local adaptation across the environmental gradient even in a small geographic range [63], fine-scale seed zones should be useful for this species.

For the second option, the novel seed transfer guidelines of *A. sachalinensis* can be predicted to effectively improve tree growth. Adaptation to WinSR and MSD are critical factors, as indicated in Figure 4 and previous studies [29,32]. Therefore, distant transfer over the climatic gradient, such as the transfer between the heavy snowfall region in the western part of Hokkaido and region with lower snowfall and more abundant sun in winter in the eastern part of Hokkaido should be avoided. Alternatively, transfer to a warmer climate is a relevant option because the increase in WI at planting sites from that in seed sources contributed to improving tree growth. Indeed, the model prediction demonstrated that some non-local seed sources have the potential to show increased tree height when at different planting sites (Figure 6). Transferring seed sources derived from S and EN into the SS region and from W into EN are candidates for increasing height growth. Among these candidates, seed sources in the S regional group for using SS is recommended because this follows the adjacent and warmer transfer.

Recently, prediction of survival and growth of trees under future climate becomes important [62], although we focused on the evaluation of current seed zones and seed transfer guidelines in this study. In fact, MRF enables us to predict future height growth across Hokkaido by applying the model to future climate scenarios. These data must be useful for mitigation and adaptation to the climate change.

## 5. Conclusions

The application of MRF to obtain results from provenance trials was found to be relevant for improving the seed transfer guidelines of *A. sachalinensis*. MRF revealed that each provenance had a different selection regime against climatic factors. Particularly, solar radiation in winter was the most important explanatory variable affecting tree height. The optimum map enables the fine tuning of seed sources for a given planting site to improve timber production. As a result, height growth was predicted to be improved using optimum seed sources rather than local seeds at some planting sites. Thus, seed source deployment can be recommended in some localities to improve timber production. However, silviculture practice requires a long time to yield harvests, and thus approaches for mitigating the risk of maladaptation of deployment materials during cultivation should be considered.

**Supplementary Materials:** The following are available online at http://www.mdpi.com/1999-4907/11/10/1058/s1, Table S1: List of seed zones and regional groups of *A. sachalinensis* in Hokkaido, Japan, Table S2: Climatic conditions of testing sites, Figure S1: Relationships between observed and predicted tree heights, Figure S2: Predicted tree heights at 10 years after planting in each provenance region, Figure S3: Difference of predicted tree height between using optimum and local seed sources, Figure S4: Averaged survival rate (left panels) and tree height with standard deviation (right panels) of transplants for each of the regional groups at each test site, Code S1: The code for developing a multivariate random forests (MRF) in R.

**Author Contributions:** Conceptualization, I.T. and W.I.; methodology, I.T. and W.I.; formal analysis, I.T. and W.I.; Visualization, I.T.; Investigation, W.I.; writing—original draft preparation, I.T. and W.I.; writing—review and editing, I.T., W.I., K.K., H.T., S.G. All authors have read and agreed to the published version of the manuscript.

**Funding:** This research was funded by Grants-in-Aid for Scientific Research from the Japan Society for Promotion of Science (JSPS KAKENHI), grant numbers 16H02554 and 20H03021.

**Acknowledgments:** We thank the staff at HRO and Hokkaido for the establishment of provenance tests and field measurements, M. Kuromaru for helpful suggestions for the analysis, and Y. Hasegawa for database construction.

**Conflicts of Interest:** The authors declare no conflict of interest.

## Abbreviations

The following abbreviations are used in this manuscript:

| | |
|---|---|
| HRO | Hokkaido Research Organization |
| WI | Warmth Index |
| TMC | Monthly mean daily minimum temperature |
| PRS | Precipitation during summer |
| MSD | Maximum snow depth |
| WinSR | Solar radiation in winter |
| SprSR | Solar radiation in spring |
| SumSR | Solar radiation in summer |
| AutSR | Solar radiation in autumn |
| MRF | Multivariate random forests |
| RF | Random forests |
| QGIS | Quantum Geographic Information System |

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
