# Peer review of "Ten Years of Provenance Trials and Application of Multivariate Random Forests Predicted the Most Preferable Seed Source for Silviculture of Abies sachalinensis in Hokkaido, Japan"

_forests, doi:10.3390/f11101058_

Round 1

Reviewer 1 Report

The authors use several provenance trials and a multivariate random forests algorithm to map predicted height growth of Abies schalinensis from different seed planning zones across Hokkaido. These analyses reveal that local seed sources are often the best performing, but sometimes non-local sources are predicted to result in improved growth, and genotypes from one seed zone in particular (S region) show universally high performance across many seed zones. These results can provide an important tool for management of this species, and the high-resolution spatially explicit maps would make it particularly relevant to both local-scale planning and evaluation of current operational practices.

COMMENTS

1) It isn’t clear to me why the authors refer to their method as a “species distribution model” (SDM) and emphasize this throughout the paper. Typically, a species distribution model will predict the geographic distribution of a species. This is not what was done; instead, the authors predict height growth from climate in a geographic framework. I thought perhaps the authors applied a common SDM algorithm for a novel purpose, but it also appears this is not what was done. Instead, the authors used multivariate random forests, which seems to be a general statistical algorithm with a wide variety of uses, not specifically a type of SDM. I think the term SDM might be misused; perhaps the method could be described as a “spatially explicit model” instead? Or, if the authors truly did make use an existing SDM method, then this should be clarified and explained.

2) Given that some of the provenance trials suffered mortality due to snow damage and canker disease, how is this likely to affect the predictions of the present study? The discussion mentions that further studies and considerations are needed, but this is rather vague. Should the current study be considered to mostly measure ideal genetic potential for height growth in a carefully managed plantation situation, or does it mostly measure realized height growth after already considering all other abiotic and biotic risks? Is there any indication whether particular provenances were more or less susceptible to damage or mortality, and could it affect the predictions? Could you clarify the preliminary findings mentioned on Line 300 where you considered mortality and confirmed the results – if you have already calculated the models including mortality why are these not also presented? This seems relevant because some provenances, for example, might be adapted to lower pest and pathogen pressure or lower risk of snow damage, and moving them to a new climate could increase this risk, even if height growth of surviving trees is improved (is there any indication of this?).

3) I was expecting to see more discussion about how these results are affected by climate change, given the existing body of literature about how climate change is expected to disrupt local adaptation. There is only a brief discussion on L335-340. For example, could the statistical model you built be projected to future climates, as a way of evaluating how height performance is expected to change in the future? I’m not suggesting this analysis needs to be done in the current paper, but it could be a straightforward suggestion for follow-up. In general, I think it’s also worth mentioning that the current results and recommendations might not be valid in changing climates. Or, perhaps they are affected by changing climates already. For example, when you found that in some cases a non-local seed source is best, does that indicate the populations have evolved local maladptation? Or could it suggest that local adaptation was optimal in the past, but the amount of climate change that has occurred already has created a mismatch between genotype and current climate?

4) The physiological importance of WinSR (winter solar radiation) is difficult to imagine. I like the interesting suggestion on L244-250 that perhaps WinSR indicates potential for photosynthesis is very early and very late winter. (Although, wouldn’t this also strongly depend on temperature?). Alternatively, is it possible that WinSR is not actually important for height growth, but it is correlated with some other, unmeasured variable that is more directly important? (For example, low WinSR could indicate high cloud cover, perhaps meaning high precipitation and more mild winter temperatures? Or something else?).

5) Fig. 5. A suggestion: I think it could be interesting to have a third map that shows the difference between the predicted height for the optimum seed provenance and the predicted height for the local provenance, i.e., b - a. This could allow the reader to more easily see how much gain can be expected by switching to using a non-local seed source, and could help facilitate discussion whether the amount of predicted gain outweighs the risks of using non-local seed in a given area.

6) L29. Not only afforestation, but any type of tree planting.

7) L84-103. Great! Very clear justification and easy to follow how you defined the seed zones based on previous practice, genetic differentiation, and climatic differences.

8) L123. Is “monthly mean daily minimum temperature” the mean over all 12 months, or is it the minimum monthly value of the coldest month? The description sounds like a mean of all 12 months, but then this would not be a measure of extreme coldness. Furthermore, the absolute values of TMC in Fig. 2 appear to be the value for the coldest month only. Please clarify the wording.

9) L131. I found the word “gaps” confusing in the paper. Is “gaps” the accepted way to describe the differences in climate? If it is not standard wording, perhaps “climatic distance” would be less awkward than “climatic gaps”.

10) Fig. 3. How do we interpret the x-axis scale? Is the line supposed to be a range of possible values, like a minimum and maximum? Or is this a bar chart? Also, the x-axis should start at zero so that the relative importance is not distorted to overemphasize the differences between variables.

11) L337. Does the S region have additional characteristics that make it suitable for wide-range planting. For example, high genetic diversity (perhaps if glacial refugia were located in the south)?

GRAMMAR

L4. fine-scale > fine scale

L12. prediction of the tree height > prediction of tree height

L13. ; > :

L58. ranges from Sakhalin, southern Kuril Islands, and Hokkaido, the northernmost island > includes Sakhalin, the southern Kuril Islands, and Hokkaido, the northernmost island

L96. “Meanwhile, it had not been any breeding material in SS”. I did not understand what this sentence is trying to say.

L140. no “s” is needed on “precipitation”

L155. an WI > a WI

L162. was 2.4-fold that in A34. This is because of that tree height > was 2.4-fold that in A34. Tree height

Fig. 6. show fraction of areas which each seed provenance > show the fraction of areas for which each seed provenance

L228. Genetic differentiation along with environmental variables > Genetic differentiation along environmental gradients

L281. which is common characteristic > which is a common characteristic

L300. preliminary > preliminarily

L316. genetic differentiations > genetic differentiation

L337. For > for

Author Response

Dear reviewer 1,   Thank you for your valuable and helpful comments. We revised our manuscript according to your suggestions. Revisions were shown in red color in the text. Please note that some revisions on English language and style were included in the text with no mark after English editing by professional. Please see the attachment PDF file to check our responses to your comments.   We hope that this revised manuscript is acceptable your publication in Forests.   Sincerely, -- Ikutaro Tsuyama Ph.D Senior researcher   Forest dynamics and diversity group, Hokkaido Research Center, Forestry and Forest Products Research Institute 7 Hitsujigaoka, Toyohira-ku, Sapporo, Hokkaido, Japan 062-8516   Phone: +81 115 90 5522 Email: itsuyama@affrc.go.jp ---------------------------------------

Reviewer 2 Report

Dear authors,

I found your work interesting for tree breeding community. Species distribution models have been currently utilized globally forest tree species included. However, your work shows the results in a new context of an important local species.

I still recommend a minor revision as I believe some improvements could be made in methodology and overall language quality. I also suggested a few corrections in terminology. Some of the technical terms you used may be confusing for potential readers.

Materials and methods: The applied model could be described in a simplified algebraic form, which would enable all readers to disentangle the individual components. Besides, please, consider submitting R code for random forest predictions as I believe some of the readers might be interested. Generally, any similar R package is a black box when presented in your current way.

General suggestions: 

line 4: good growth - please use better wording throughout the text (production, growth performance?)

line 6: substitute smaller height growth - awkward terminology

line 7: use seed sources instead of seeds

line 16: substitute north by northern; adjective in this context

line 62: I suggest using the term stem straightness

line 94: plus trees instead of mother trees

line 95: probably a term founders population would be more suitable instead of breeding materials

line 115: omit then and omit "but a complete dataset" - confusing

Figure 4. "Rugs" on x-axes are missing within the bottom figures (SprSR)

line 287-line 292: I appreciate that you included the explanations for the canker occurrence. However, it points to the major methodological question - whether these plots should have been included in your study, after all? I understand that these sites are necessary for the overall prediction, but we see there is a strong bias induced.

line 310: My question is - have you thought about including future climatic data in your study? I think you should at least comment on this topic.

Author Response

Dear reviewer 2,

Thank you for your valuable and helpful comments. We revised our manuscript according to your suggestions. Revisions were shown in red color in the text. Please note that some revisions on English language and style were included in the text with no mark after English editing by professional. Please see the attachment PDF file to check our responses to your comments.

We hope that this revised manuscript is acceptable your publication in Forests.

Sincerely,
--
Ikutaro Tsuyama Ph.D
Senior researcher

Forest dynamics and diversity group, Hokkaido Research Center,
Forestry and Forest Products Research Institute
7 Hitsujigaoka, Toyohira-ku, Sapporo, Hokkaido, Japan 062-8516

Phone: +81 115 90 5522
Email: itsuyama@affrc.go.jp
---------------------------------------
